# Floral Aroma and Pollinator Relationships in Two Sympatric Late-Summer-Flowering Mediterranean Asparagus Species

**DOI:** 10.3390/plants12183219

**Published:** 2023-09-10

**Authors:** Leonardo Llorens, Joan Tomàs, Pere Ferriol, María Trinitat García, Lorenzo Gil

**Affiliations:** 1Interdisciplinary Ecology Group, Department of Biology, University of the Balearic Islands (UIB), Ctra. Palma-Valldemossa Km. 7.5, E-07122 Palma, Balearic Islands, Spain; pere.ferriol@uib.es (P.F.); lorenzo.gil@uib.es (L.G.); 2Department of Biology (Botany), University of the Balearic Islands (UIB), Ctra. Palma-Valldemossa Km. 7.5, E-07122 Palma, Balearic Islands, Spain; joantomas1@gmail.com; 3Scientific and Technical Services, University of the Balearic Islands (UIB), Carretera de Valldemossa Km. 7.5, E-07122 Palma de Mallorca, Balearic Islands, Spain; trinidad.garcia@uib.es

**Keywords:** *Asparagus*, floral scent, volatile compounds, plant–pollinator interaction, flowering synchrony, gynodioecy

## Abstract

This research delves into plant–pollinator relationships within the Mediterranean region, focusing on two synchronous and sympatric asparagus species: *A. acutifolius* and *A. albus*. For the first time, the floral scents of the genus Asparagus are reported. We investigate the volatile organic compounds (VOCs) present in their floral scents and their impact on pollinator attraction. Captured flower-emitted VOCs underwent solid-phase microextraction of headspace (SPME-HS) and gas chromatography and mass spectrometry (GC-MS) analysis. The investigation confirms distinctive aroma profiles for each species. A. albus predominantly emits benzene derivatives and sesquiterpenes, while *A. acutifolius* is characterized by carotenoid derivatives, monoterpenes, and sesquiterpenes. The only shared compounds between the two species are the sesquiterpenes (*Z*,*E*)-α-farnesene and (*E*,*E*)-α-farnesene. A positive correlation links peak floral aroma intensity (benzenoids in *A. albus* and ionones in *A. acutifolius*) with a higher pollinator visit frequency, emphasizing the critical role of intense floral scents in pollinator attraction. The study of reproductive aspects reveals almost complete gynodioecy in *A. acutifolius*, influencing unique dynamics for the two species. These adaptations hold significant importance within the Mediterranean ecosystem, particularly during the late dry summer period, when a limited number of plant species vie for a shared primary pollinator.

## 1. Introduction

Plants produce many volatile organic compounds (VOCs), which frequently play important ecological roles in plant–plant, plant–insect, and plant–microorganism interactions. It is known that most entomophilous pollination interactions are linked to visual and olfactory attractants. A specific floral scent may contribute to reproductive isolation among closely related species (e.g., [1,2]). Most plants emit a common core group of compounds [3], but these sometimes exhibit quantitative and qualitative variability depending on temporal or environmental variations [4,5]. Some phylogenetic studies have recognized this characteristic of variation in floral volatiles as useful for constructing phylogenies at a distinct level, such as that of the clade [6], genus [7], or species [8]. Often, taxa are not characterized by specific volatiles, although those with some degree of plasticity are found within at least some of the same generative lineages. Furthermore, the characteristic fragrance of each species is determined by a broad spectrum of VOCs [9], and the quantitative and qualitative variability in these volatile compounds can be specific to each plant species [10]. Therefore, each taxon’s “floral aroma” can be used in its taxonomic characterization [11].

On the other hand, flowering phenology [12,13] and the singular complex blend of different volatile compounds in each plant play key roles in biological competition to attract a certain pollinator species [14] at a precise moment. In addition, the floral scent may influence the flower constancy and the abundance of pollinators [15,16]. Ensuring effective pollen transfer contributes to reproductive success and the maintenance of reproductive barriers among species [17,18].

Wild *Asparagus acutifolius* L. and *Asparagus albus* L., which are members of the Asparagaceae family, are characteristic species that thrive in maquis and shrublands found on the rocky soils of the arid Mediterranean region. They exhibit a coexisting pattern and flower synchronously within Southwestern Europe and Northwestern Africa [19]. These species possess a rich history of utilization for their culinary and medicinal purposes, with their spears being locally consumed. Moreover, they hold value for apiculture (*Apis mellifera* L.) and possess diuretic properties [19,20,21,22]. Furthermore, they are recognized as potential new crops that are suitable for cultivation in semi-arid Mediterranean environments [23,24]. *A. acutifolius* predominantly blooms in late summer—specifically, from August to September—thus aligning with the arrival of cooler temperatures and the initial late-summer rains. However, during extended periods of drought, occasional flowering and fruiting can occur in spring, albeit less frequently. In contrast, *A. albus* initiates its flowering phase in late summer regardless of rainfall, and it abstains from spring blooming.

The sex of the two species of plants is different; *A. albus* has only hermaphroditic flowers, while the combinations reported in *A. acutifolius* plants are diverse. Some sources indicate that it is a dioecious species [25,26], while other authors consider it to be gynodioecious, with plants having female (unisexual by abortion) or only hermaphroditic flowers [27,28]. The gynodioecious phenomena described extensively in angiosperms have been linked to an evolutionary mechanism to avoid inbreeding [29,30,31].

To achieve successful pollination, entomophilous plants must produce signals that effectively stimulate insect sensitivity. Volatile aromas are a common and effective floral trait linked to attracting and selecting pollinators. The response of pollinators in terms of type and intensity can vary depending on the specific combination or individual chemical compounds present [32,33]. Thus, interactions between floral traits and pollinators have been described as “floral scent emission and pollination syndromes” [34,35,36,37]. Consequently, floral characteristics often provide valuable insights into the most likely pollinator, and the chemical profile of a floral aroma can aid in identifying the animal groups—primarily invertebrates—that act or have acted as the main pollinators [38,39]. Furthermore, there is often a close relationship between the timing of floral scent emission and the periods of pollinator activity, as floral scent emission tends to occur synchronously with the activity of pollinators [40,41].

In contrast to the extensive information on the biology or genetics of the genus *Asparagus* (e.g., [25,28,42,43,44]), knowledge of other aspects is more scarce. This is the case for volatile organic compounds (VOCs), the study of which has been limited to the chemical composition of the shoots that are used as food [45,46,47] or to the relationship of volatiles from vegetative parts with insect pests [48,49]. Currently, there are no references on the volatiles of *Asparagus* flowers or on the role that they may play in pollination. There is also scarce information on the VOCs of congeneric, sympatric, and synchronous flowering species.

The main objective of this study is to analyse and evaluate the interspecific differences in the chemical profiles of floral volatile organic compounds (VOCs) in two Mediterranean wild asparagus species. Additionally, this study aims to investigate the floral sexuality of both species and establish a correlation between aroma production and pollinator behaviour during the flowering period.

## 2. Results

### 2.1. Flowering Phenology

The flowering patterns of *A. albus* and *A. acutifolius* are characterized by highly synchronous flowering. The flowering timing of *A. albus* and *A. acutifolius* is reported in Figure 1. The flowering intensities of the two species were different; *A. acutifolius* produced 85% of its flowers in about 9 days, whereas *A. albus* did so in about 27 days. In both populations, *A. albus* flowered 10–15 days earlier than *A. acutifolius*. Furthermore, the flowering of *A. albus* lasted between 33 and 39 days, whereas the flowering of *A. acutifolius* only lasted between 12 and 24 days. Additionally, hermaphroditic *A. acutifolius* plants produced approximately 52.6% more flowers than the female plants.

### 2.2. Pollinator Responses and Reproductive Success

#### 2.2.1. Pollinators

The pollinator observations showed that in *A. albus*, over 66% of the visits were by *A. mellifera* (Hymenoptera: Apidae), 28% were by the genus *Lasioglossum* (Hymenoptera; Halictidae) (two morphotypes), and the remaining 6% were by rare or non-pollinator visitors, such as *Chrysoperla mediterranea* (Hölzel 1972), Diptera, or other Hymenoptera (Figure 2, Table 1). In *A. acutifolius,* the most common visitors were also *A. mellifera*, accounting for 96% of pollinator visits, and the remaining 4% were *Lasioglossum* (one morphotype) and an undetermined fly (two morphotypes). In all of the cases analysed, bees visiting the flowers of the two *Asparagus* species carried pollen of this genus as their main load (Table 2).

#### 2.2.2. Reproductive Success (Breeding System)

In *A. acutifolius*, the reproductive success in naturally pollinated female plants was 23.4%, and it was 62% in the manually cross-pollinated plants (Table 3). In addition, the reproductive success in hermaphroditic plants was low in all cases, including in the manual cross-pollination tests (between 0.5 and 2.5%). In *A. albus*, the fruit set percentages were 34.2% in natural conditions, 60% with manual cross-pollination, and between 19.0 and 20.0% in assays related to autocompatibility (isolated open pollination, natural self-pollination, and manual self-pollination).

### 2.3. Compounds of Floral Scents

A total of fifty VOCs were identified in the flower emissions of *A. acutifolius* and *A. albus* (Table 4). The components were monoterpenes, carotenoid derivatives, sesquiterpenes, aromatic benzenoids, phenylpropanoids, and alkanes.

The main volatile compounds in *A. acutifolius* were monoterpenes, carotenoid derivatives, and sesquiterpenes. Among monoterpenes, the most abundant compound was (*E*)-geranyl acetone, and geranyl acetone, geranyl acetol, and sulcatone were identified in smaller amounts. The apocarotenoids α-ionone, β-ionone, and dihydro-β-ionone were present (average concentration >65%), and the sesquiterpene hydrocarbons (*Z*,*E*)-α-farnesene and (*E*,*E*)-α-farnesene were also present. The VOCs in *A. albus* were represented by sesquiterpenes (*Z*,*E*)-α-farnesene and (*E*,*E*)-α-farnesene, aromatic benzenoids, and phenylpropanoids, including benzaldehyde, phenylacetaldehyde, phenethyl acetate, benzyl alcohol, and phenethyl alcohol. In addition, small amounts of alkanes were identified for this species. The sesquiterpenes (*Z*,*E*)-α-farnesene and (*E*,*E*)-α-farnesene were common to both species, but in different proportions (mean = 0.39 and 10.77 in *A. acutifolius* and mean = 3.06 and 19.35 in *A. albus,* respectively).

ANOSIM analysis showed that the interspecific difference in volatile composition was greater than the intraspecific difference (R = 1, *p*-value = 0.0001).

SIMPER analysis showed that the most influential compounds that accounted for up to 80% of the differences between the aroma profiles of the two species were benzeneacetaldehyde (35.9%), which was present only in *A. albus*, and dihydro-β-ionone, β-ionone, and (*E*)-geranyl acetone, which were only found in *A. acutifolius* (Table 5, Figure 3).

### 2.4. Volatile Emission Timing

Low qualitative or quantitative differences in the emission patterns of VOCs were detected throughout the day for *A. acutifolius* (33–34% monoterpenes; 33–34% sesquiterpenes; 32–24% carotenoid derivatives), while *A. albus* showed a dynamic emission pattern throughout the daily sampling hours (Table 6). The aromatic benzenoids and phenylpropanoids increased in abundance throughout the day (from 20% at 08:00 UTC to 63% at 18:00 UTC). The sesquiterpenes remained constant with a percentage close to 30%, and the alkanes, which comprised 46% at 08:00 UTC, decreased throughout the day until they disappeared in the last sample of the day.

On the other hand, there were differences in the amounts of volatiles produced during the day, with *A. albus* producing a greater quantity of VOCs than *A. acutifolius* at any time of the day. Additionally, slight differences in the time of maximum production existed, with *A. albus’* maximum occurring between 06:00 UTC and 08:00 UTC and *A. acutifolius*’ maximum taking place around 11:00 UTC (Table 6).

### 2.5. Volatile Emission and Pollinators Visits

For the two Asparagus species, the Pearson’s product–moment correlation between scent production and number of visits is positive, statistically not significant, and very large (r = 0.60, 95% CI (0.60, 0.97), t(3) = 1.29, *p* = 0.289 for *A. acutifolius*, and r = 0.77, 95% CI (−0.34, 0.98), t(3) = 2.11, *p* = 0.125 for *A. albus*).

## 3. Discussion

### 3.1. Phenology, Breeding System, and Sexuality

The capacity for coexistence among species sharing the same habitat requires, among other factors, a series of interactions that are often associated with defence and success in attracting common pollinators. In this process, the circadian clock plays a fundamental role in enabling plants to adapt and respond to environmental changes [50]. In particular, the synchronization of flowering among each species becomes a determining factor for achieving successful fruiting. This becomes especially relevant when plants develop in locations with highly contrasting seasonal water and climatic conditions, exhibit self-incompatibility, or coexist with other species that possess similar floral characteristics. Phenological monitoring suggests that the flowering of the two *Asparagus* species is primarily determined by two regulators: the photoperiod in *A. albus* and water availability (precipitation) following a period of water stress in *A. acutifolius*, which occurs in the Mediterranean region after the driest period of summer. As a result, intense flowering occurs within short time periods (*A. acutifolius* produces 85% of its flowers in 6 days, and *A. albus* does so in 27 days). These floral displays create a high concentration of floral scent that, along with other floral cues—such as colour, the white patches of flowering *A. albus* plants in the arid landscape at the end of August and beginning of September, and rewards—promotes the attraction and concentration of pollinators, thereby favouring fruit sets [51,52,53].

The flowers and plants of *A. albus* are hermaphroditic, which is consistent with the traditional descriptions, and the fruit set results of natural self-pollination and manual self-pollination show that the species is partially autogamous. However, there is heterogeneity in the information regarding the sexual characteristics of the flowers of *A. acutifolius*. While some authors [27,28] consider the plants to be gynodioecious, with hermaphrodite and female individuals coexisting in the same population, others [25,26] describe them as a dioecious species. Our observations have revealed the presence of two types of flowers: female flowers, which lack pollen production (aborted anthers) (see Appendix A), and others that possess both pollen and reduced pistils, albeit with varying degrees of reduction (see Appendix A). The fertility of the latter is extremely low, with estimated pollination rates reaching a maximum of 0.3%. Individuals with these flowers can be classified as either hermaphrodites or, based on their functionality, nearly unisexual males. These floral characteristics align with a tendency towards dioecy in this species. On the other hand, hermaphroditic plants produced a greater number of flowers than female plants (52.6% more). This finding is also consistent with the theory of sexual selection in dioecious plants [54,55], which suggests that male plants invest more in attracting pollinators than females do, as observed in the case of *A. acutifolius*.

In female plants of *A. acutifolius*, the differences in the reproductive success values between manually cross-pollinated flowers (62.0%) and naturally pollinated (23.4%) flowers indicated a deficient pollination performance. This can be attributed to several factors, including excessive distance between hermaphroditic (male) and female plants, an inadequate proportion of sexual individuals in the populations, a low number of pollinators, or the fact that combined visits to flowers of both sexes do not allow for a sufficient supply of pollen to the female flowers. Based on our results, the latter scenario appears to be the most likely explanation.

### 3.2. Pollinators and Floral Scents

*Apis mellifera* is the main pollinator of the two *Asparagus* species under study. The frequency of its visits and its pollen transport capacity are determining factors for the reproductive success of both species. Moreover, in *A. albus*, we also observed a notable number of visits by insects belonging to the genus *Lasioglossum*. It is also well-established that bees utilize not only visual, but also olfactory floral signals to locate appropriate host plants (Appendix A). Additionally, numerous studies have recognized the correlation between the intensity and quality of floral scent and the provision of floral rewards (see, for example, [56,57,58]).

In both species, the greatest production of volatiles occurred in the early morning hours. On the other hand, there was a positive relationship between the intensity of aroma production and the number of pollinator visits. This relationship was particularly consistent in *A. albus*, which exhibited the highest concentration of benzenoids during this time of day.

Most compounds emitted by *A. albus* flowers were benzenoids, with phenylacetaldehyde being the main compound (almost 65% of total emission). Some authors [59] proved that the floral signal phenylacetaldehyde was positively correlated with the amount of reward in *Brassica rapa* L., which is a species that is mainly pollinated by *A. mellifera* and *Bombus terrestris* L. The hydrocarbon group of sesquiterpenoids ((*Z*,*E*)-α-farnesene and (*E*,*E*)-α-farnesene) was common in both species of *Asparagus*. The ecological role of these sesquiterpenes has been reported as attractive to the honeybee [60,61], bumblebees, *B. terrestris* [2] butterflies, and moths [62,63]. The waxy sweet fragrance of pentadecane, despite being an unusual floral compound and having a lower load in the floral scent emissions of *A. albus* (1.5% during the early hours of the day), has been described in some Orchidaceae species [64] and in the floral volatiles of cacao trees [65]. In turn, the floral bouquet of *A. acutifolius* was dominated by compounds derived from the degradation of carotenoids, with dihydro-β-ionone being the major volatile compound, representing almost 40% of the total scent emissions. This ionone has been reported as a floral volatile in various species, is frequently associated with more widespread ionones, and is known to play an essential role in pollinator attraction [66,67,68]. β-Ionone has been found to attract males of orchid bees, *Euglossa mandibularis* (Hymenoptera, Apidae) [69], to repel the cabbage butterfly (*Pieris rapae*) [70], and to inhibit the movement of the crucifer flea beetle (*Phyllotreta cruciferae*) [71]. Additionally, ionone compounds have been described as having a repellent effect against the spider mite *Tetranychus urticae* [72], one of the most important pathologies in the *Asparagus* genus. Aromatic monoterpenes, such as geranyl derivatives (about 20% of the total scent), were emitted by *A. acutifolius* and are found in many plants’ essential oils [73,74]. Geranyl acetone has been shown to play an important role in plant–pollinator interactions, to have antibacterial effects against many bacterial strains, and to exhibit antioxidant activity [74]. Thus, the volatile profiles of the two species, especially that of *A. albus*, conformed to the attraction–information function appropriate to the sensitivity of their main pollinators. Furthermore, the substantial presence of ionones in the floral scent of *A. acutifolius* suggests that, apart from their role in attracting honeybees, they may also serve as a defensive and deterrent mechanism against other insects.

The study provides evidence that two coexisting species with synchronous flowering and distinct sexual profiles (hermaphroditic and functional dioecious) are able to attract the same primary pollinators by exhibiting contrasting floral profiles. Consequently, both species can successfully bear fruit and coexist. As observed in other sympatric species, the diversification of floral scents could play a significant role in maintaining species balance within the vegetation [75]. However, it remains to be determined how variations in aroma comparably influence the behaviour of pollinators in relation to other floral or phenological features.

## 4. Materials and Methods

### 4.1. Study Site

During August and September of 2016, 2019, and 2022, samples of wild populations of two species belonging to the genus *Asparagus*—*A. acutifolius* (subgenus *Asparagus*) and *A. albus* L. (subgenus Protoasparagus)—were simultaneously studied in their natural habitats in Majorca (Balearic Islands, Western Mediterranean). For *A. acutifolius*, the localities studied were Alaró, Estellencs, Palma (Genova and UIB-Campus), and Sóller. For *A. albus*, the studied plants were from Andratx, Artà, Campos, Palma (Genova and UIB-Campus), and Llucmajor (Figure 4).

In the two locations of Palma (Genova and UIB-Campus), the populations of the two species are mixed, while in the others, the plants thrived under similar soil and climatic conditions. All of these localities have a Mediterranean climate and are characterized by mild temperatures (average annual range between 16.5 and 17.8 °C) and irregular rainfall (average annual range between 320 and 600 mm), which primarily occurs in autumn and spring. The summers are hot and dry. From a bioclimatic point of view, the localities of Alaró, Artà, Estellencs, and Sóller fall into the Thermo-Mediterranean dry belt, while Andratx, Campos, Llucmajor, and Palma fall into the Thermo-Mediterranean semiarid belt [76].

### 4.2. Flowering Phenology

To determine the phenology of *A. acutifolius* and *A. albus*, in 2019 and 2022, six plants of each species were monitored in two localities: Palma—Genova (39°33′30.95″ N, 2°36′50.76″ E) and UIB-Campus (39°38′09.90″ N, 2°38′30.96″ E). For each plant, five branches were randomly tagged, and the numbers of open flowers were counted. Flowering was considered to have started when the first flower was observed to be open and ended when each plant no longer possessed any flowers with anthers [77]. This monitoring was performed every 3 days at the same time of day—early in the morning.

### 4.3. Breeding System

For both species, to examine the breeding system and the contributions of visiting insects and floral organs to pollen transport and fructification, seven experimental treatments were conducted on the plants in their natural habitats. (1) Open pollination in 10 natural populations: In each population, 10 flowers were marked on 5 plants of each species (a total of 500 flowers) before anthesis and then allowed natural visits by insects. (2) Isolated open pollination: In this case, a similar procedure to that in the previous case was used. However, the assays were conducted at 5 locations per species (the Llucmajor population was not monitored) by using four-year-old cultivated plants in each of them. Before anthesis, 20 flowers were marked on each plant (a total of 100 flowers) and allowed to be naturally visited by insects. The remaining tests were carried out on 5 populations at 1 plant per population and 10 flowers per plant. (3) Spontaneous self-pollination: Flowers that were about to open were selected and marked. Each of them was bagged with 1 × 1 mm nylon nets until withering. (4) Manual self-pollination: A procedure similar to that in the previous case was used, but the flowers were bagged before anthesis, and the newly opened flowers were hand-pollinated with their own pollen and then bagged again. (5) Manual cross-pollination: A procedure similar to that in the previous case was used, but the flowers were bagged prior to anthesis, and the newly opened flowers were hand-pollinated with pollen collected from other individuals at least 10 m away from the pollen. (6) Removal of anthers: Before anthesis, flowers were emasculated and then left to be visited by insects. (7) Removal of the anthers and bagging: the same procedure as that in the previous case was used, but the flowers were kept enclosed in nets with 1 × 1 mm holes until withering. Due to the gynodioecious character of *A. acutifolius*, tests 2, 3, 4, 6, and 7 were not performed on functionally female plants. In each assay, the number of fruits produced was determined.

### 4.4. Scent Sampling and Analysis and Identification of Compounds

For this purpose, individuals from the 11 populations were used for each species (5 for *A. acutifolius* and 6 for *A. albus*). Floral volatile samples were collected by using the headspace solid-phase microextraction-sampling technique (HS-SPME). Compounds were extracted in a manual SPME holder with 10 mL glass vials and polydimethylsiloxane–divinylbenzene (PDMS-DVB) fibres with a film thickness of 100 μm (Supelco Inc., Bellefonte, PA, USA), and they were conditioned according to the instructions provided by the manufacturer. To determine the VOCs emitted by the flowers of the two species, many samples were collected between 08:00 and 11:00 UTC (hours with the highest number and diversity of visitors) in 3 plants per population. Immediately after harvesting, 15 flowers of each plant were placed in a 10 mL glass vial. Then, to equilibrate the analytes, these vials were placed in a water bath at 25 °C for 20 min. Afterward, the SPME fibre was exposed in the sealed vial for 30 min at 25 °C to adsorb the analytes [78]. Furthermore, a mixture of a certain number of precisely weighted saturated n-alkanes (C7–C30) (Merck, Rahway, NJ, USA) diluted with hexane (w = 5%) was used as a standard in the vials [79]. To determine the variations that may have existed in their emissions throughout the day, a similar methodology was used, with 10 flowers per vial collected in 3 plants of the UIB population at 05:00, 08:00, 11:00, 14:00, and 17:00 UTC. For these determinations, nonyl acetate (Sigma-Aldrich, St. Louis, MO, USA) was used as the internal standard.

The scent samples were analysed by using an Agilent 6980 GC-MDS 5975 inert XL system (Agilent Technologies, EUA). Chromatographic separation was performed by using a Supelcowax 10 capillary GC column (60 m × 0.25 mm × 0.25 μm) with helium as a carrier gas at a flow rate of 1.3 mL min^−1^. For thermal desorption, PDMS-DVB fibre from the SPME was aged for 3 min in the inlet of the GC. The analytical conditions used were as follows: spitless injection at 220 °C and ionization occurring via electron impact (70 eV; source temperature 180A °C). The initial oven temperature was programmed as follows: 45 °C for 2 min followed by an increase to 250 °C at a rate of 5 °C min^−1^, which was kept for 5 min. These settings were enough for the quantitative desorption of all analytes studied. The MS scan range was set between 45 and 300 uma. The total ion chromatograms and mass spectra were processed by using the Turbomass version 5.1 software (Perkin-Elmer, Inc., Waltham, MA, USA).

Chromatographic peaks were identified by comparing the mass spectra with the NIST 14 library (similarity > 80%) and with published data (NIST, http://webbook.nist.gov/chemistry/ accessed on 1 July 2020; PubChem, http://pubchem.ncbi.nlm.nih.gov/ accessed on 1 July 2020), by comparison with their Kovats retention indexes relative to saturated alkanes C7–C30 (Merck), and by comparison with an internal database. The main compounds were also verified by comparison with standards (Sigma-Aldrich).

### 4.5. Pollinator Monitoring

Insect visits to flowers were assessed by analysing recordings obtained with Bushnell HD cameras. Observations took place in August, once per week, on days with weather stability; they were conducted between 06:00 and 20:00 UTC in branches with 20 flowers from 3 plants for each monitoring location. Nocturnal observations were not performed because these flowers remain closed at night. During weekly sampling, five quantifications were performed for each plant for 10 min every 3 (−2) h (at 05:00, 08:00, 11:00, 14:00, and 17:00 UTC), for a total of 900 min of observation (150 min in each location). The floral visitors were considered pollinators if they clearly contacted the reproductive parts of the flowers and had pollen grains of *Asparagus* fixed on their bodies based on the field observations and pollen control in brightfield microscope specimens [80]. To confirm the pollen-carrying capacity of the visiting insects, the following procedure was implemented. Initially, the insects were captured with a net and carefully transferred into a vial equipped with a plastic grid at one end. By gently guiding the insects with a small stick, they were directed into the plastic net. To extract the pollen, a small piece of glycerol–gelatine held in place with tweezers was employed. The gelatinous globules containing pollen were subsequently placed onto a microscope slide, which was heated and covered with a glass cover slip. These samples were preserved until further examination. After successfully extracting the pollen, the insects were released back into their natural habitat. Optical microscopy was used to identify the pollen samples by comparing them with pollen from our own collections and with pollen described in specific references [81,82]. Samples of each pollinator species were captured for identification by local specialists.

### 4.6. Statistical Analysis of the Data

Analyses of floral scent differences between the species were performed with the R version 4.1.2 statistical software (R Project for Statistical Computing) by using the vegan library [83]. Compounds present in less than 10% of the samples and in amounts of less than 1% were excluded from the analysis. To assess similarities between individual samples, Bray–Curtis similarity indices were calculated and arranged in a triangular matrix. The similarity matrix was then used to perform ANOSIM analyses [84]. To identify the percentage contributions of each volatile compound to the mean dissimilarity between the two Asparagus species, similarity percentage analysis (SIMPER, the simper function in vegan) was used [85]. From the volatile composition of the samples, a logarithmically transformed data matrix was prepared and analysed by using detrended correspondence analysis (DCA) (decorana function, vegan package). Furthermore, to visualize the aromatic composition of all samples and their grouping by species, a heat map of the average hierarchical linkage clustering based on the Bray–Curtis dissimilarity was plotted. To explore whether the fruiting percentage of female Asparagus acutifolius plants varied with distance to the nearest hermaphrodite, Spearman’s rank correlation coefficient was calculated. To examine the relationships between scent production and Apis mellifera visits in both Asparagus species, a Pearson correlation was calculated.

## Figures and Tables

**Figure 1 plants-12-03219-f001:**
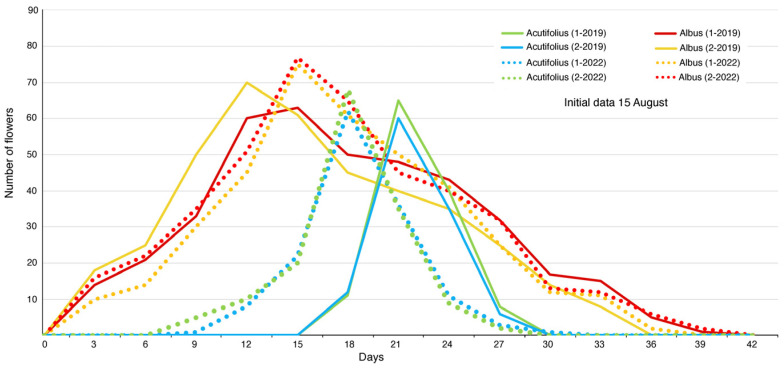
The flowering timing, in 2019 and 2022, for two populations of *A. albus* and *A. acutifolius*.

**Figure 2 plants-12-03219-f002:**
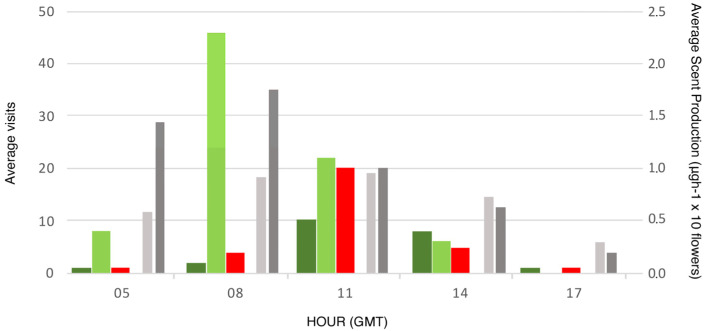
Total scent production (total VOCs broadcasting timetable) and an average number of visits to *Asparagus acutifolius* (females) and *A. albus* flowers along the day (GMT hours). Dark green: *A. acutifolius*—*Apis mellifera*; light green: *A. albus*—*A. mellifera*; red: *A. Albus*—*Lasioglossum* sp.; Grey columns: average scent production (10 flowers, *n* = 10), in μgh^−1^. Light grey: *A. acutifolius*; dark grey: *A albus*.

**Figure 3 plants-12-03219-f003:**
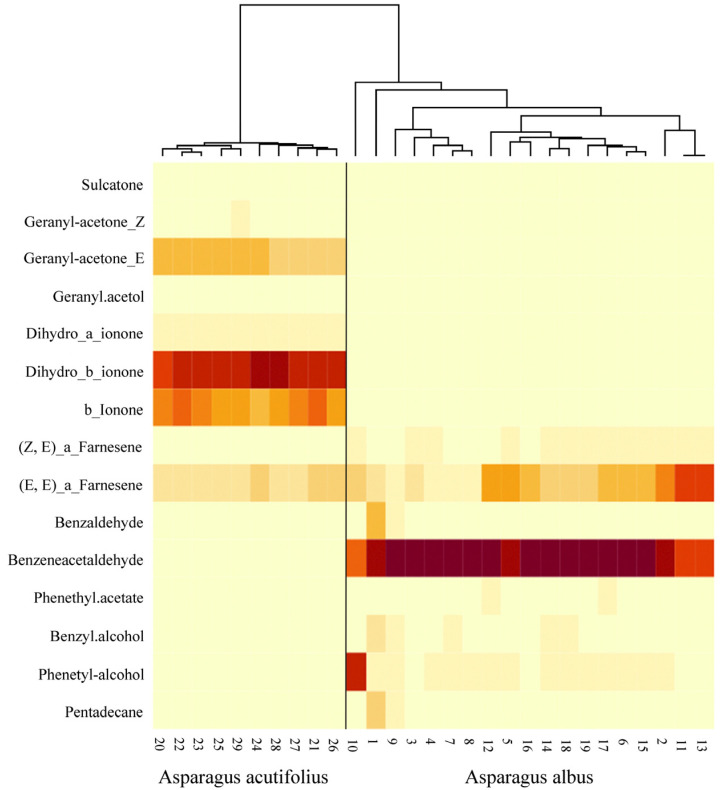
Similarity dendrogram of *Asparagus acutifolius* and *A. albus* VOCs. Colours intensity is related to the level of presence of component in each sample.

**Figure 4 plants-12-03219-f004:**
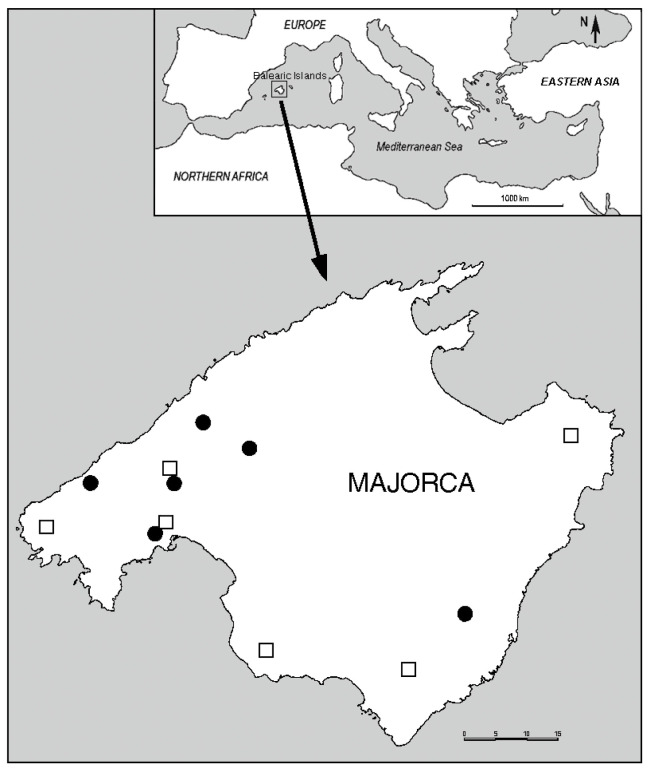
Location of the Asparagus populations included in the study. *A. acutifolius*: circles; *A. albus*: open squares.

**Table 1 plants-12-03219-t001:** Average number and minimum–maximum values of *Apis mellifera* visits in bloom at *Asparagus acutifolius* and *A. albus* for different observation times.

Species	Hour (UTC)				
	05	08	11	14	17
*A. acutifolius* (females)	1 (0–2)	2 (1–4)	10 (7–14)	8 (3–10)	7 (0–2)
*A. acutifolius* (hermaphrodites)	2 (1–2)	7 (4.9)	9 (6–15)	3 (1–6)	1 (0–1)
*A. albus*	8 (5–10	46 (32–62)	22 (18–25)	6 (2–8)	0

**Table 2 plants-12-03219-t002:** Percentage of conspecific pollen on bees captured on flowers of *Asparagus acutifolius* (Estellencs) and *A. albus* (Santanyi). N = number of captures.

	*A. acutifolius* (Females)		*A. acutifolius* (Hermaphrodites)		*A. albus*	
Year	% ± s.d.	N	% ± s.d.	N	% ± s.d.	N
2016	77.6 (6.73)	9	96.7 (2.25)	6	96.6 (3.38)	10
2019	68.7 (7.79)	10	93.2 (3.19)	5	98.1 (1.07)	7
2022	80.0 (4.93)	11	96.50 (1.41)	8	95.5 (1.52)	6

**Table 3 plants-12-03219-t003:** Fruit set of *Asparagus acutifolius* and *A. albus* under different treatments, in % (sd). 1: natural open pollination (*n* = 500); 2: isolated open pollination (*n* = 500): 3: natural self-pollination (*n* = 50); 4: manual self-pollination (*n* = 50); 5: manual cross-pollination (*n* = 50); 6: pollination anthers removed (*n* = 50); 7: pollination anthers removed and bagged (*n* = 50).

Species	Treatment						
	1	2	3	4	5	6	7
*A. acutifolius* (females)	23.4 (2.3)				62.0 (5.4		
*A. acutifolius* (hermaphrodites)	0.4 (1.3)	0.3 (0.2)	0.3 (0.2)	0.2 (0.4)	1.2 (1.4)	1.1 (1.5)	0
*A. albus*	54.2 (6.9)	19.0 (3.2)	10.0 (1.6)	11.0 (2.5)	60.0 (4.2)	18.6 (2.4)	0

**Table 4 plants-12-03219-t004:** Volatile organic compounds (VOCs) of Asparagus acutifolius (n = 10) and A. albus (n = 19). Occurrences (in %) refer to the total number of samples that contained this compound; Mean: mean composition of floral compounds, expressed as a relative abundance. Identification method: a = Kovats retention index; b = mass spectrum; c = confirmed with authentic compound; occurrence in %; SD: standard deviation. * = VOCs commonto the two species.

Compound	Identification	Kovats Index	*Asparagus acutifolius*		*Asparagus albus*	
			Mean ± SD	Occurrence	Mean ± SD	Occurrence
**Monoterpenes and relatives**						
Sulcatone (= 6-Methyl-5-hepten-2-one)	a,c	1340	0.6 ± 0.81	40		
(*Z*)-Geranyl acetone	a,b	1420	3.07 ± 0.6	100		
(*E*)-Geranyl acetone	a,b,c	1870	15.89 ± 2.4	100		
6,10-Dimethylundeca-5,9-dien-2-ol)	a	1454	1.18 ± 0.18	100		
**Carotenoids derivatives**						
Dihydro-α-Ionone	a	1795	4.96 ± 0.87	100		
Dihydro-β-Ionone	a,b,c	1854	38.25 ± 2.64	100		
β-Ionone	a,b,c	1938	24.45 ± 2.92	100		
**Sesquiterpenes**						
(*Z*,*E*)-α-Farnesene *	a,b,c	1701	0.39 ± 0.5	40	3.06 ± 2.15	79
(*E*,*E*)-α-Farnesene *	a,b,c	1740	10.77 ± 1.16	100	19.35 ± 12.5	100
**Benzene derivatives**						
Benzaldehyde	a,b,c	1512			1.63 ± 3.93	37
Phenylacetaldehyde	a,b,c	1595			63.85 ± 16.23	100
Phenethyl acetate	a,b	1734			0.78 ± 1.75	26
Benzyl alcohol	a,b,c	1866			1.86 ± 2.26	63
Phenethyl alcohol	a,b,c	1902			5.54 ± 9.38	95
**Alkanes**						
Pentadecane	a,b,c	1504			1.42 ± 3.1	37

**Table 5 plants-12-03219-t005:** SIMPER analysis. Average and cumulative contributions of most influential compounds. AvA: average in *Asparagus acutifolius*; AvB: average in *A. albus*; Cumsum: cumulative contribution (in %).

VOC	AvA	AvB	Cumsum
Phenylacetaldehyde	63.85	0	35.9
Dihydro-β-Ionone	0	38.25	57.4
β-Ionone	0	24.46	71.1
(*E*)-Geranyl acetone	0	15.89	80.0
(*E*,*E*)-α-Farnesene	19.35	10.77	86.6
Phenethyl alcohol	5.54	0	89.7
Dihydro-α-Ionone	0	4.96	92.5
(*Z*)-Geranyl acetone	0	3.07	94.5
(*Z*,*E*)-α-Farnesene	3.06	0.39	95.8
Benzyl alcohol	1.86	0	96.9
Benzaldehyde	1.63	0	97.8
Pentadecane	1.42	0	98.6
Geranyl acetol	0	1.18	99.2
Phenethyl acetate	0.78	0	99.7
Sulcatone	0	0.60	100

**Table 6 plants-12-03219-t006:** Temporal scent production and distribution of volatile compounds in *Asparagus acutifolius* and *A. albus* (in %). Scent production (10 flowers) in μgh^−1^.

	VOC	Hour (UTC)				
		05	08	11	14	17
*Asparagus acutifolius*	Alkanes	33.05	33.24	33.01	13.70	12.34
	Carotenoids derivatives	33.56	33.87	32.81	56.53	63.12
	Monoterpenes	33.32	32.78	33.10	26.78	24.01
	Scent production: media (max/min)	0.6 (0.8–0.4)	0.8 (1.0–0–7)	0.9 (1.1–0.5)	0.7 (0.9–0.6)	0.3 (0.1–0.4)
*Asparagus albus*	Alkanes	31.41	44.92	20.41	0	0
	Benzene derivatives	36.75	20.36	44.48	63.10	75.02
	Sesquiterpenes	31.56	33.7	34.20	36.76	24.97
	Scent production: media (max/min)	1.4 (1.6–1.2)	1.7 (1.9–1.5)	1.0 (1.1–0.6)	0.6 (0.9–0.3)	0.2 (0.3–0.0)

## Data Availability

The raw data supporting the conclusions of this article will be made available by the authors, without undue reservation.

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
