# Peer review of "Floral Aroma and Pollinator Relationships in Two Sympatric Late-Summer-Flowering Mediterranean Asparagus Species"

_plants, 2023, doi:10.3390/plants12183219_

Round 1

Reviewer 1 Report

In this study, authors aimed to analyze the interspecific differences in floral volatile in two wild asparagus species and establish a correlation between aroma production and pollinator behavior. Although the designing and results of this paper is reasonable. The writing and formatting is pretty rough. 

1 The writing and formatting of this paper is unsatisfactory. The general quality of figure and table production is low (like the missing of y axis, standardization of unit et al)

2 Tables should be editable (not pictures).

3 Without Figure legends for Fig2-5, it’s difficult to accurately understand the indication of the Figure. Detailed legends should be provided.

4 Except the visiting frequency evaluation, Y tube assay should be employed to directly compare the attraction of floral volatile of these 2 species to the pollinator.

5 Correlation between the visitation frequency and volatile profile should be further established by person analysis or PLS analysis.

 The writing and formatting of this paper is unsatisfactory, which should be substantially improved.

Author Response

Dear Reviewer. We are grateful for your comments and suggestions on the manuscript. They have been helpful and I hope that the modifications we have made in response to your suggestions will help to improve the manuscript. 

Comments and Suggestions for Authors

 1 The writing and formatting of this paper is unsatisfactory. The general quality of figure and table production is low (like the missing of y axis, standardization of unit et al).

The text has been reviewed by specialized professional staff and modified accordingly. The tables have been modified and reformatted. Thus, in table 2 the Y-axis legend has been added. The units and their standardization have also been revised (like scent production in μgh-1,  time timetable in GMT).   

The original Figure 4 has been removed and the values that allowed it to be obtained are shown in Table 6.

2 Tables should be editable (not pictures).

The tables are included in editable format. They were originally sent as .xlsx files and as pictures.

3 Without Figure legends for Fig2-5, it’s difficult to accurately understand the indication of the Figure. Detailed legends should be provided.

Originally the figure captions were sent in a dedicated file. They have now also been included in the main file.

4 Except the visiting frequency evaluation, Y tube assay should be employed to directly compare the attraction of floral volatile of these 2 species to the pollinator.

We wholeheartedly concur with your observation. Nevertheless, it is important to note that the primary objective of this study did not involve a comparison of the attractiveness of various scents. Furthermore, given that the key volatile compounds in both scents are known to attract the established pollinators, a definitive determination in this regard was not within the scope of our research.

5 Correlation between the visitation frequency and volatile profile should be further established by Pearson analysis or PLS analysis.

The Pearson correlation has been calculated. The results are presented in:

1 Results: Lines 191-195 (2.5)

1 Method: Lines 420-421

For the two Asparagus species, the Pearson's product-moment correlation between scent production and number of visits is positive, statistically not significant, and very large (r = 0.60, 95% CI [-0.60, 0.97], t(3) = 1.29, p = 0.289 for A. acutifolius, and r= 0.77, 95% CI [-0.34, 0.98], t(3) = 2.11, p = 0.125 for A. albus)

3 Discussion (3.2). Lines 249-253

Comments on the Quality of English Language

6 The writing and formatting of this paper is unsatisfactory, which should be substantially improved.

The manuscript has been reviewed and correct now by MDPI experienced native English speaking editor.

Reviewer 2 Report

Explanation of the Fig. 2,3,4 are not given! So I cannot review and value the paper.

Nothing is said on the human perception of the floral aroma.

Belong the perhaps pristine pollinating Lasioglossum to different species? Are there 2 or 3 species involved. Are the 2 Asparagus pollinated by different Lasioglossum spp.?

Author Response

Dear reviewer. We are grateful for your comments and suggestions on the manuscript. They have been helpful and I hope that the modifications we have made in response to your suggestions will help to improve the manuscript. 

Comments and Suggestions for Authors

1-3 Comments in attached pdf.

 1- The comments indicated in the pdf on the italic formatting of two plant names (A. albus) have been changed.

 2- The text "Volatile aroma are a common and effective floral trait linked to attracting and selecting specific pollinator types. (lines 72, 73 of the original) has been replaced by "Volatile aromas are a common and effective floral trait linked to attracting and selecting pollinators" (lines 76,77 of the corrected file).

 3- The question stated in Table 2: "Does this mean that e.g. 77.6 % of the pollen load was from A. acutifolius?

Correct. As indicated in the explanation of the table, the values refer to pollen of the same species (conspecific pollen).

 Other Comments

4- Explanation of the Fig. 2,3,4 are not given! So I cannot review and value the paper.

Nothing is said on the human perception of the floral aroma.

Originally, the figure captions were sent in a dedicated file. They have now also been included in the main file.

 5- Belong the perhaps pristine pollinating Lasioglossum to different species? Are there 2 or 3 species involved. Are the 2 Asparagus pollinated by different Lasioglossum spp.?

We can confidently affirm that two morphotypes have been observed in A. albus (line 117), and one in A. acutifolius (line 120). The one from A. acutifolius coincides with one from A. albus.

 The manuscript has been reviewed and correct now by MDPI experienced native English speaking editor.

Round 2

Reviewer 1 Report

The manucript was well revised. No further comments。